# Clinical characteristics and serotype association of dengue and dengue like illness in Pakistan

Najeeha Talat Iqbal [1]*, Kumail Ahmed[2], Aqsa Khalid[2], Kehkashan Imtiaz[3], Qamreen Mumtaz Ali[1], Tania Munir[2], Syed Faisal Mahmood[4], Unab Khan[2], Badar Afzal[2], Farah Naz Qamar[2], Jesse J. Waggoner[5], Hannah Fenelon[6], Helene McOwen[7], Erum Khan[3], Peter Rabinowitz[6], Wesley C. Van Voorhis[7]

1 Department of pediatrics and child Health, Department of Biological and Biomedical Sciences, Aga Khan University, Karachi, Pakistan, 2 Department of pediatrics and Child Health, Aga Khan University, Karachi, Pakistan, 3 Department of Pathology and Laboratory Medicine, Aga Khan University, Karachi, Pakistan, 4 Department of Medicine, Aga Khan University, Karachi, Pakistan, 5 Department of Medicine, Division of Infectious Diseases, Emory University School of Medicine, Atlanta, Georgia, United States of America, 6 Department of Environmental/Occupational Health Sciences, Family Medicine, Global Health, University of Washington, Seattle, Washington, United States of America, 7 Center for Emerging and Re-emerging Infectious Diseases (CERID), University of Washington, Seattle, Washington, United States of America

* najeeha.iqbal@aku.edu

## Abstract

### Background

Pakistan has been an endemic country for dengue virus since 1994, with a significant increase in cases reported in 2022 largely due to heavy rainfall and flooding. All four serotypes of the dengue virus (DENV) are present in Pakistan, with DENV 1 and DENV 2 being the most prevalent. The current study aims to explore the clinical presentations and features of dengue fever in a tertiary care hospital.

### Methodology

We enrolled and studied 349 cases of suspected and confirmed dengue presenting for care at the Aga Khan University Hospital in Karachi between June 2021 and November 2023. Collected data on cases included clinical symptoms and laboratory results including qRT-PCR and serotype characterization.

### Findings

The majority of subjects enrolled (75%) had mild disease without warning signs, while 11% exhibited warning signs, 1.4% had severe dengue, and 12.6% had no dengue diagnosis. Patients with severe dengue (SD) had significantly higher levels of liver enzymes (AST and ALT) compared to those with non-severe dengue (NSD) (AST; p = 0.024 and ALT; p = 0.047). Additionally, a higher grade of thrombocytopenia was significantly associated with hospitalization (p = 0.0008), and prolonged illness (p = 0.03). Both Platelet (p < 0.0001) and WBC counts (p = 0.001) were significantly

**Data availability statement:** All relevant data are within the paper and its Supporting Information files.

**Funding:** This work was supported by the National Institutes of Health, CREID Network [United World for Antivirus Research Network (UWARN)], NIH/NIAID (U01AI151698 to WVV); and American and Asian Centers for Arboviral Research (A2CARES), NIH/NIAID (U01 AI151788). The study design and data collection protocols were developed as part of the United World for Antiviral Research Network (UWARN), one of the NIH-funded Centers for Research in Emerging Infectious Disease (CREID). The funders had no role in study design, data collection and analysis, decision to publish, or preparation of the manuscript.

**Competing interests:** The authors have declared that no competing interests exist.

lower in dengue PCR-positive patients in comparison to Dengue PCR-negative. Among those tested for dengue serotypes, DENV 1 (34%) and DENV 2 (45%) emerged as the predominant serotypes, with mixed infections accounting for 17%. The sensitivity of q-RT PCR was found to be 87.25% and the specificity of 68.35%. qRT-PCR detected 43.5% of cases with viral fever initially screened negative by IgM or NS1.

## Conclusion

The epidemiology of dengue fever during a widespread outbreak in 2022 showed a predominance of DENV 1 and DENV 2 serotypes with milder phenotype of viral illness. Screening with rapid tests requires further confirmation by molecular assay in cases with dengue and dengue-like illness. The sensitivity of q-RT PCR using gold standard.

## Author summary

Dengue (DENV) is a mosquito-borne disease endemic in Pakistan, with seasonal outbreaks often triggered by monsoon rains and flooding. In this study, we describe the clinical characteristics of laboratory-confirmed Dengue patients and those with Dengue-like illness using qRT-PCR. Molecular diagnostics detected a significant proportion of cases with diverse clinical presentations, underscoring the limitations of conventional screening tests. Our findings highlight the evolving clinical spectrum of Dengue, which is frequently missed during initial screening. Accurate clinical decision-making remains challenging in the absence of highly sensitive diagnostic tools. Strengthening viral diagnostic modalities is essential for improving patient management and disease surveillance. Among the detected serotypes, DENV-2 was the most prevalent in non-severe Dengue cases, particularly during monsoon-driven outbreaks in Pakistan.

## 1. Introduction

Dengue is prevalent in tropical and subtropical regions and is transmitted by Aedes aegypti mosquitoes [1]. While most cases are self-limited, some can progress to the life-threatening hemorrhagic shock syndrome. Dengue is caused by four distinct serotypes of the dengue virus: DENV 1, DENV 2, DENV 3, and DENV 4 [2]. A single infection provides relative protection against reinfection with that specific serotype, whereas reinfection with other serotypes may lead to more severe illness [3] All four serotypes have been reported in different outbreaks of dengue infection in Pakistan with DENV 2 and DENV 3 to be the most prevalent serotypes [2,3]. The distribution of dengue serotypes in Pakistan is uneven, with DENV 2 being reported in 1994 and subsequently in 2005 and 2006, along with DENV 3 [3]. DENV 1 and DENV 2 were identified as the most prevalent serotypes during

the 2022 outbreak in Pakistan [4]. Studies conducted in Dengue endemic countries like, Indonesia, and Malaysia also reported the severity of the dengue infection association with co-infection of DENV 1/ DENV 2 and concurrent infection of DENV 2 and DENV 3 [5,6].

In 2022, torrential monsoon rains accompanied by flooding affected many villages and households in Pakistan, with rain fall 175% higher than the annual average over the past 60 years, and as high as 726% of the annual average in Sindh province. This severe event caused damage to crop, livestock, and infrastructure and had a huge toll on the communities with low socio-economic status. These flood conditions were accompanied by an upsurge in waterborne and vector-borne diseases, including a rapid spread of dengue infection[7,8]. The World Health Organization (WHO) report on the 2022 dengue outbreak in Pakistan, concluded that there was a total of 25,932 confirmed cases and 62 fatalities among individuals seeking medical assistance for health concerns [9].

The Aga Khan University Hospital in Karachi, Pakistan is a tertiary care center that receives cases of dengue from the Karachi metropolitan area; a highly urbanized city with suboptimal sanitation and weak infrastructure of society, open garbage, and leaky pipelines that provide an ideal breeding environment for vectors. The fact that all four Dengue serotypes were circulating during the outbreak of 2022 provided an opportunity to compare clinical aspects of the serotypes receiving care in the same facility.

It is also important to acknowledge the possibility of other circulating viruses like Zika virus and Chikungunya viruses which share a common host (human) and mosquito vectors (primarily *Aedes aegypti*) and have similar biological and ecological factors leading to epidemiological synergy [10,11].

We therefore reviewed cases of confirmed and suspected dengue that were hospitalized or treated as out-patients between 2021–2023, which captured the period of the 2022 outbreak, allowing us to highlight the clinical features and circulation of the different dengue serotypes, as well as the sensitivity of PCR compared to clinical diagnosis.

## 2. Methodology

### 2.1. Ethics statement

Ethical approval was obtained from the Ethical Review Committee (ERC) of Aga Khan University prior to the initiation of the study with approval number ERC#4794. Written informed consent was obtained from all participants or legally authorized representatives. In the case of children, parental consent or assent was obtained as appropriate.

### 2.2. Participant enrollment

The protocols were designed to create a cohort of patients presenting with clinical symptoms consistent with acute febrile illness (suggestive of Arboviral or other emerging viral infection). The sample size was determined by availability of data during the data collection process which was started in June 2021 and continued during the year 2023, but data was included in this manuscript was collected until November 2023. Details of 349 samples are mentioned in S1 Appendix. The majority of subjects belonged to the Karachi region which is one of the most densely populated areas of Pakistan (Mean population density 21862). **Fig 1** shows the flow of subject enrollment.

To define cases for enrollment as confirmed or suspected dengue, we followed clinical symptoms and classification as per World Health Organization (WHO) and Pan American Health Organization (PAHO) guidelines (PAHO World Health Organization, 2017) [12]. The criteria include (a) Admission within 1–2 weeks and clinical suspicion of arbovirus infection or confirmed diagnosis of arboviral disease, (b) Sudden, moderate to high-grade fever (>39° C/ 102.2° F), typically of 2–7 days duration, (c) any manifestations such as nausea or vomiting, diarrhea, exanthema (non-characteristic, appears between days 5–7), mild to intense pruritus, myalgia, headache, abdominal pain, petechiae/skin bleeding, and dental/gum bleed, (d) patient's age between 1–75 years of age.

Patients not meeting inclusion criteria and those refusing consent, out of age criteria were excluded from the study.

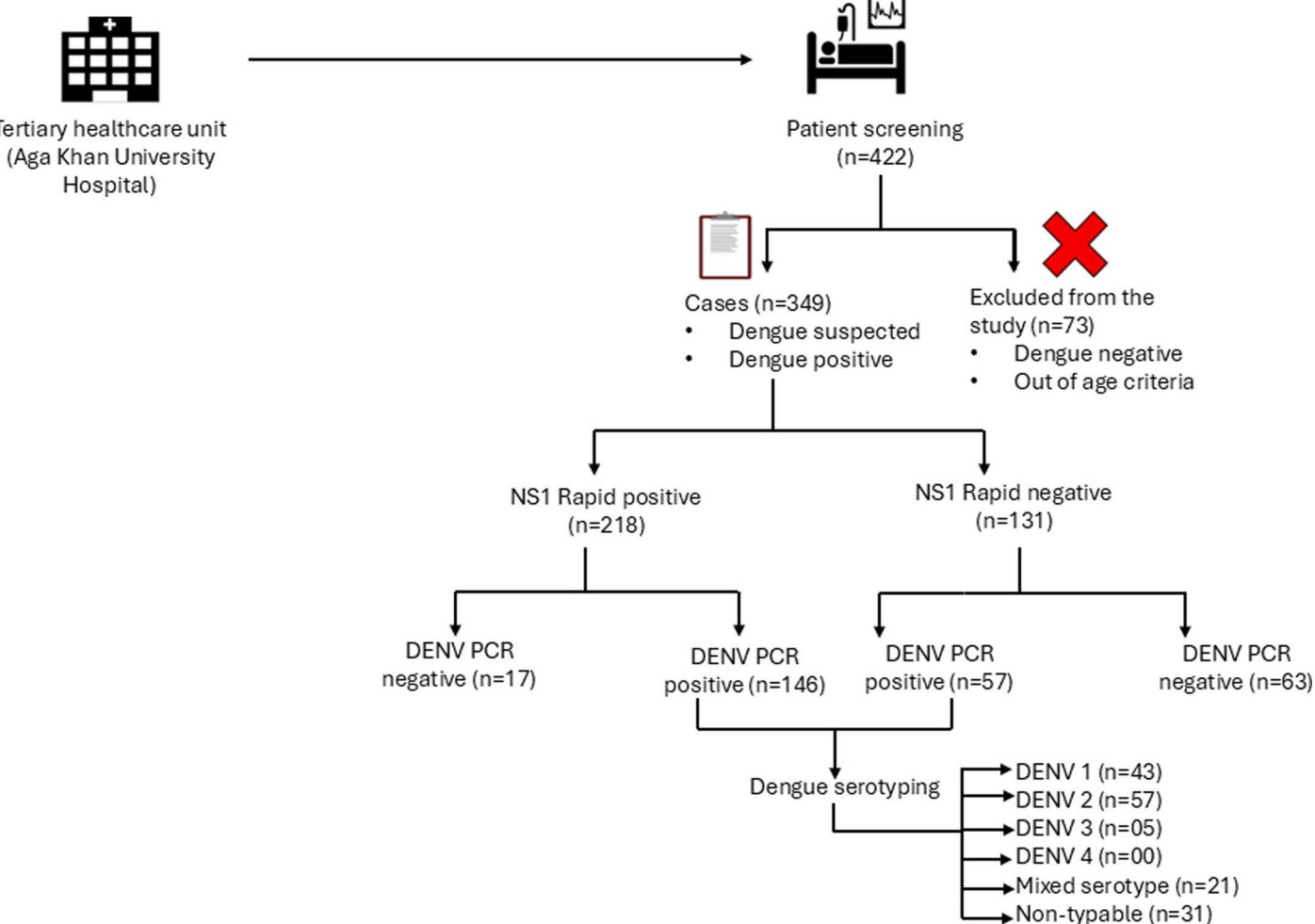

**Fig 1. Enrollment and dengue Screening workflow in UWARN study.** Enrollment and Dengue Screening workflow in UWARN study. Flow chart of recruitment of dengue cases in UWARN study, Pakistan site. Fever of other origins were not enrolled in our study (n = 73). The figure has been made using open source software clipart https://openclipart.org/.

## 2.3. Case definition

The enrolled patients were classified into DENV+ and DENV- based on serological and molecular evidence of DENV.

**Dengue Positive**: Participant with NS1+ or IgM+ or PCR+ (n = 287).

**Dengue Negative:** Participant with negative NS1− or IgM− and/or PCR− (n = 62).

## 2.4. Clinical diagnosis

A total of 349 Dengue patients were classified on dengue severity as per WHO criteria into severe dengue (SD; n = 5) and non-severe dengue (NSD), which was further divided into dengue with warning signs (DWS; n = 39), dengue without warning signs (DWOS; n = 261), and with other diagnoses (n = 43) One sample did not have a record available for classification.

## 2.5. Sample collection

Blood samples (n = 349) were collected in an EDTA tube from each participant enrolled during the acute phase of the disease. Samples were transported to the laboratory within 30 minutes of collection and tested for NS1 antigen using Abbott

Panbio dengue Early Rapid Kit at the time of enrollment followed by sample storage at 4 °C. Plasma was separated and stored at −80 °C.

## 2.6. RNA extraction

RNA was extracted from the plasma sample using the QIAamp Viral RNA Extraction Kit (Qiagen-GmbH, Hilden, Germany) following the manufacturer's protocol. The RNA was eluted in elution buffer and stored at −80 °C. The integrity of the extraction reagents and the successful recovery of RNA from clinical samples was confirmed bythe presence of RNase P using RNase P probe-based qRT-PCR assay from Biosearch Technologies.

## 2.7. qRT-PCR amplification

Extracted RNA was subjected to the Pan dengue and dengue serotyping PCR using the Platinum Superscript III Invitrogen One-Step qRT-PCR kit (Invitrogen, Life Technologies, Carlsbad, CA, USA) [13]. The Pan dengue and serotyping primers and probes were purchased from Biosearch Technologies (Tables A-C in S1 Text). The positive controls for DENV1, DENV2, DENV3, and DENV4 were obtained from BEI resources, and were used at 1:10 dilution . All primers and probes working concentrations were adjusted at $20\,\mu M$ followed by the addition of RNA template into each reaction. qRT-PCR was performed using CFX96 Bio-Rad (Hercules, CA) platform. The cycling conditions included an initial step at 52°C for 15 minutes followed by 94°C for 2 minutes for the reverse transcriptase reaction and initial denaturation, respectively. The PCR was set up for 45 cycles with denaturation at 94°C for 15 seconds and annealing with acquisition at 55°C for 20 seconds, followed by extension at 60°C for 20 seconds, and a final extension at 68°C for 20 seconds. Fluorophore signals were detected on their respective channels. Any curve crossing the threshold before 42 cycles was considered positive after baseline and threshold adjustments.

## 2.8. Statistical analysis

Descriptive statistics (Mean ±Standard Deviation) were calculated for quantitative variables. Laboratory values in the clinical setting were assessed in comparison to the normal reference range specific to the corresponding gender as required. For the DENV PCR + and PCR- samples, Wilcoxon rank-sum test was performed to identify the differences among three groups. Moreover, the Kruskal Wallis test was applied to determine the differences of hematological parameters and Ct value distribution in dengue serotypes. For sensitivity analyses, ROC analysis was performed using "Final diagnosis" and "DENV PCR" results using SPSS and R software using "pROC" package. AUC was calculated for specificity and sensitivity and confidence intervals (CIs). P value < 0.05 was used as a cutoff for statistical significance. All the analysis was performed using R software v.4.2.2.

## 3. Results

### 3.1. Demographics and clinical features of dengue infection

We screened 422 patients from in-patient and out-patient settings of Aga Khan University. Of those, 349 subjects met the inclusion criteria (Fig 1). Cases with confirmed dengue diagnosis and subjects with dengue-like symptoms (Dengue suspected) were enrolled during the study period (2021 – 2023). Approximately half of the cases (n = 168) were enrolled during Sep-Oct 2022 due to heavy rainfall and floods in Sindh province (Fig 2A). The enrollment pattern depicts that most cases (n = 85) DENV+ from NS1 were included from Sep-Oct 2022 (Fig 2B). Initially 218/349 (62.4%) patients were tested positive for NS1 antigen. The mean age of DENV positive (Participant with NS1 + or IgM + or PCR+) was 33 ± 14 years and DENV negative (NS1-or IgM- and/ or PCR-) were 31 ± 15 years, including both Male (n = 209) and Female (n = 140).

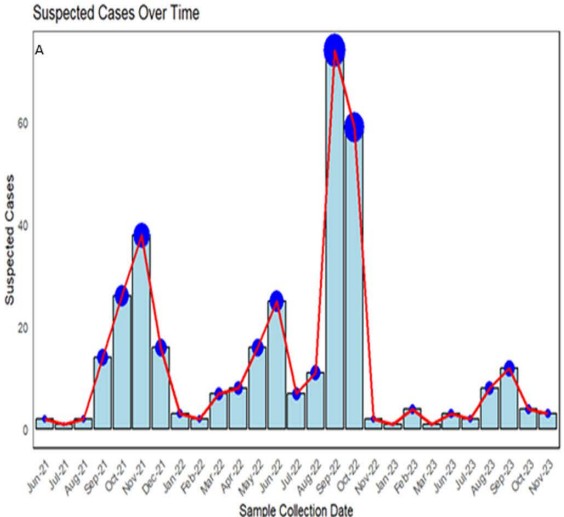
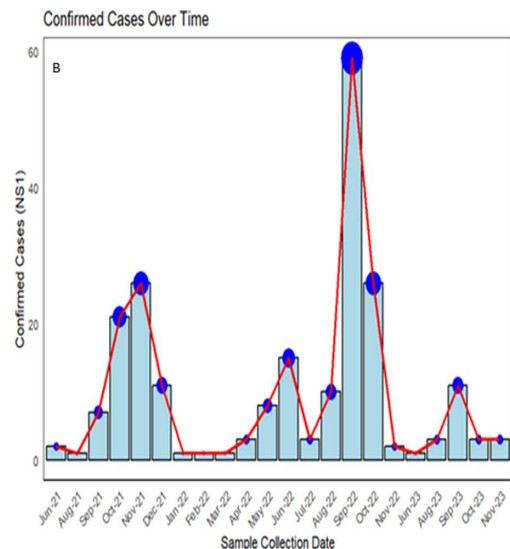

**Fig 2. Enrollment of patients in the study over a period of two years.** An epidemiological curve showing the number of Suspected (A) and confirmed dengue cases (B) during 2021-2022 according to the diagnosis by NS1 antigen; red curve shows the trend of confirmed dengue cases.

Male gender (60%) was found to be predominant in comparison to female gender. Respiration rate (p = 0.005), lymphocyte (p = 0.001), neutrophils (P < 0.001), platelet count (P < 0.001), WBC (p < 0.001), ALT (p < 0.001), AST (p < 0.001) and creatinine (p < 0.001) showed statistically significant differences between DENV positive and negative cohorts. (Table 1).

After the initial screening, Pan-Dengue-PCR was performed on 283/349 subjects. A total of 283 PCR were performed and 66 samples were not processed further for PCR analyses due to limitation of reagents. Of the 283, 203 (71%) were identified as DENV+ through qRT-PCR, and within this subset, 157 individuals were subjected to testing for four DENV serotypes (S1 Fig). The overall screening and algorithm of testing is shown in Fig 1. Among the DENV positive cases as per case definition (PCR+ OR IgM+ OR NS1+) (n = 287), male gender was predominant 172 (60%) compared to females 115(40%) (Tables 1 and D in S1 Text).

The clinical signs and symptoms, as classified by the WHO dengue severity was assessed in patients who tested positive for both NS1 and PCR. The history of fever at the time of enrollment was identified as the most common symptom across all three (98%) [dengue without warning signs (DWOS), dengue with warning signs (DWS), and severe dengue (SD)] categories of dengue infection. Moreover, higher percentages of abdominal pain, body aches and diarrhea were observed in SD infection in comparison to other categories (Fig 3).

### 3.2. Sensitivity of molecular diagnostics for dengue infection

The sensitivity of q-RT PCR was calculated based on the based on the clinician diagnosis. The sensitivity of q-RT PCR was calculated based on the case definition used for Dx positive (Dx+) cases which comprised of both NS1+/IgM+, NS1 OR IgM positive OR clinical presentation of "Dengue fever", "Dengue suspects", "Dengue like illness" and all "viral fever". This constitutes a total of 349 cases which include Dx+ (n = 262) [NS1+/IgM+/Clinical Dx+] and Dx − (n = 87) [NS1−/IgM−/Clinical Dx−] groups.

The sensitivity of q-RT PCR was found to be 87.25% [CI:81.89%-91.50%] accompanied by specificity of 68.35% [CI:56.92% to 78.37%], a Positive Predictive Value (PPV) of 87.68%[CI: 83.68% to 90.81%], and a Negative Predictive Value (NPV) of 67.50% [CI: 58.46% to 75.40%], the AUC for qRT-PCR positive against clinical case definition was in the acceptable range of 0.780 (CI: 0.714%- 0.846%), as detailed in Tables 2 and Table E in S1 Text.

**Table 1. Demographics and clinical characteristics of DENV cohort-clinical definition (N = 349).**

| Characteristic | Negative N = 62[1] | Positive N = 287[1] | p-value[2] |
|---|---|---|---|
| Age | 31 (15) | 33 (14) | 0.3 |
| Gender | | | 0.97 |
| Female | 25 (40%) | 115 (40%) | |
| Male | 37 (60%) | 172 (60%) | |
| **Vitals** | | | |
| Systolic (mmHg) | 117 (13) | 117 (15) | 0.69 |
| Diastolic (mmHg) | 76 (12) | 74 (11) | 0.33 |
| Heart Rate (bpm) | 90 (20) | 87 (20) | 0.13 |
| Temperature | 36.86 (0.64) | 37.07 (0.99) | 0.29 |
| Respiration (bpm) | 19.06 (2.58) | 18.08 (2.13) | **0.005** |
| Oxygen Saturation | 97.72 (2.74) | 97.66 (2.64) | 0.4 |
| **Hematological parameters** | | | |
| Hematocrit (%) | 37.4 (5.6) | 39.2 (5.9) | 0.067 |
| WBC Count | 7.6 (3.8) | 5.5 (3.3) | **<0.001** |
| Lymphocyte Count | 29 (17) | 37 (19) | **0.001** |
| Neutrophil Count | 61 (15) | 52 (20) | **<0.001** |
| Hemoglobin | 12.27 (2.10) | 12.88 (2.05) | 0.056 |
| Platelet Count | 221 (85) | 113 (85) | **<0.001** |
| ALT | 144 (463) | 186 (481) | **<0.001** |
| AST | 114 (331) | 244 (588) | **<0.001** |
| BUN | 14 (8) | 11 (13) | 0.052 |
| Creatinine | 0.80 (0.35) | 0.97 (0.64) | **0.032** |

[1]Mean (SD); n (%).

[2]Wilcoxon rank sum test; Pearson's Chi-squared test.

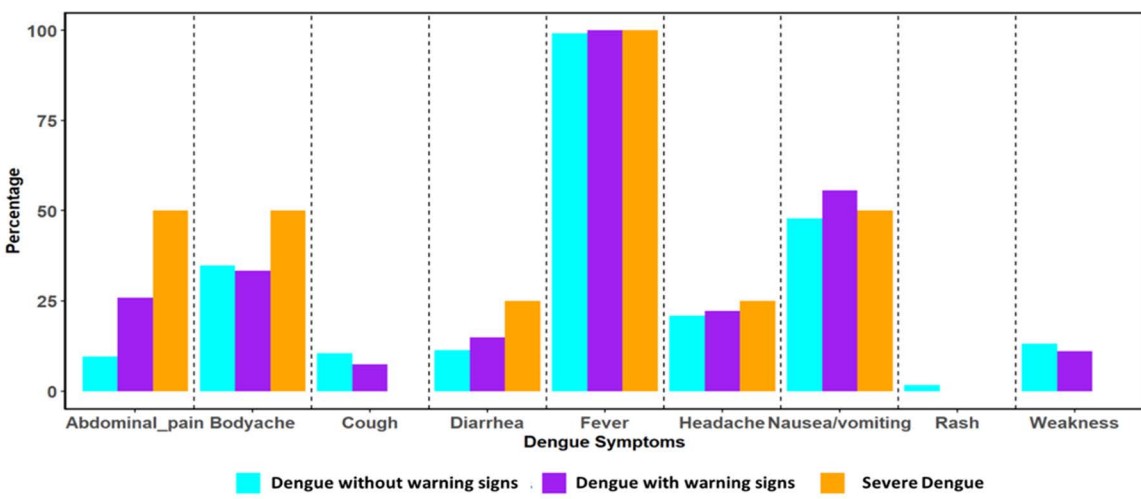

**Fig 3. Clinical symptoms in Dengue patients as per WHO criteria.** Associated symptoms of dengue infection classified according to WHO criteria. Fever is found to be the most common symptom among all categories. Orange represents SD; Cyan represents DWWS, and purple represents DWOS.

**Table 2. Sensitivity of clinical diagnosis of DENV with clinical case definition.**

| Statistics | Value (%) | 95% Confidence Interval (CI) |
|---|---|---|
| Sensitivity | 87.25 | 81.89-91.50 |
| Specificity | 68.35 | 56.92 to 78.37 |
| Positive Predictive Value (PPV) | 87.68 | 83.68 to 90.81 |
| Negative Predictive Value (NPV) | 67.50 | 58.46 to 75.40 |
| AUC | 0.780 | 0.714 to 0.846 |

In addition to the above analyses, sensitivity and specificity were also calculated for the gold standard positive samples (NS1+or IgM+) which include 267 cases excluding strong clinical suspicion of Dengue fever (n = 16). The qRT-PCR sensitivity (86.63%; CI:80.90% to 91.16%) and specificity (67.50%; CI: 56.11% to 77.55%) remained same with minor differences in AUC of 0.773 [95% CI: 0.705 to 0.840] (Tables F and G in S1 Text).

Forty-one cases presented with multiple unrelated symptoms secondary to viral illness but were neither NS1+nor IgM+for Dengue. The clinical presentations were shown in Sheet B in S1 Appendix. These samples were also subjected to q-RT PCR. We found 19 of 41 of these samples to be positive for dengue PCR and DENV1 and/or DENV2 serotypes were detected (DENV1 = 10, DENV2 = 7, DENV1/DENV2 = 2). This was an additional finding in our study which was picked by a sensitive PCR test. A total of 46% of samples were positive with q-RT PCR.

### 3.3. Serotype-specific detection of dengue virus

Of 203 qRT-PCR positive cases, 157 were subjected to DENV serotyping by PCR. The mean Ct values for DENV 1 was 31.05, DENV 2 was 32.95, DENV 3 was 33.81 and mixed serotype was 32.16. Moreover, no significant differences were observed in the mean Ct values between all the groups (DENV 1, DENV 2, DENV 3 and mixed serotypes) (S2 Fig and Table B in S1 Text). Among 157, DENV 1 (n = 43) was 27% and DENV 2 (n = 57) was 36%, such that DENV 1 and DENV 2 were the predominant serotypes. Moreover, DENV 3 accounted for only 3% (n = 5), and DENV 4 was not detected through serotype-specific PCR analysis. Additionally, 13% of cases (n = 21) were identified as mixed serotypes (S1B Fig and Sheet C in S1 Appendix). Moreover, there were also a few non-typeable samples 20% (n = 31), which were not classified using primer set.

### 3.4. Clinical presentation of dengue infection

Clinical parameters, including vitals and hematological parameters, were available for the 349 enrolled patients at the time of enrollment. The mean vitals values were similar between dengue PCR positive and negative patients except for temperature parameter (Table 3). Similarly, the majority of DENV patients exhibited mean normal body temperature, however, a proportion of patients (31% DENV+ and 16% DENV−) showed slightly raised body temperature (>37.0 °C). The observed differences were statistically significant (p = 0.013). Abnormal respirations (>20 breaths per minute) were noted in 9% of individuals with DENV+ and 6% with DENV−. Moreover, oxygen saturation was normal in all cases (i.e., 100% for DENV+ and 98.5% for DENV−). The details of the normal and deranged vitals for both groups are mentioned in Sheet D in S1 Appendix.

The hematological parameters of DENV+ and DENV− (n = 283) patients showed low platelet count, elevated AST, and deranged ALT levels (Table 3). The mean platelet counts within 1–2 days of enrollment for DENV+ patients were below the normal range of 150 – 400 x 10^9/L. Similarly, ALT and AST levels were also found to be deranged from the normal range (≤40]. The percentages of the hematological parameters and comparison with normal ranges is shown in Sheet D in S1 Appendix. Furthermore, low platelets (p < 0.001) and low WBC count (p = 0.001) were observed in DENV+ group (Table 3 and Sheet D in S1 Appendix). In addition, the differences in hematological profiles of DENV serotype specific groups (DENV 1, DENV 2, DENV 3 and mixed) were also showing statistically significant differences for Creatinine (p = 0.022)

**Table 3. Demographics and clinical characteristics of qRT-PCR confirmed cases.**

| Characteristic | Negative N = 80[1] | Positive N = 203[1] | p-value[2] |
|---|---|---|---|
| Age | 31 (1 - 70) | 34 (3 - 75) | 0.28 |
| Gender | | | 0.37 |
| Female | 30 (38%) | 88 (43%) | |
| Male | 50 (63%) | 115 (57%) | |
| **Vitals** | | | |
| Systolic (mmHg) | 118 (84 - 152) | 117 (84 - 167) | 0.45 |
| Diastolic (mmHg) | 75 (50 - 112) | 74 (45 - 109) | 0.37 |
| Heart Rate (bpm) | 89 (13 - 182) | 85 (10 - 145) | 0.3 |
| Temperature | 36.80 (36.00 - 39.50) | 37.12 (34.00 - 41.20) | **0.013** |
| Respiration (bpm) | 18.46 (15.00 - 29.00) | 18.07 (12.00 - 29.00) | 0.11 |
| Oxygen Saturation | 97.45 (84.00 - 100.00) | 97.74 (90.00 - 100.00) | 0.58 |
| **Hematological Parameters** | | | |
| Hematocrit (%) | 38.4 (22.8 - 51.4) | 38.3 (23.3 - 53.4) | 0.76 |
| WBC Count (10^9/L) | 7.1 (1.3 - 20.0) | 5.6 (1.0 - 21.9) | **<0.001** |
| Lymphocyte (%) | 36 (4 - 97) | 37 (5 - 97) | 0.43 |
| Neutrophils (%) | 54 (16 - 91) | 52 (9 - 91) | 0.34 |
| Hemoglobin (g/dL) | 12.52 (7.30 - 16.60) | 12.69 (7.80 - 17.50) | 0.64 |
| Platelet Count (10^9/L) | 186 (12 - 515) | 109 (6 - 479) | **<0.001** |
| ALT (IU/L) | 280 (10 - 5,241) | 163 (9 - 3,515) | 0.34 |
| AST (IU/L) | 165 (21 - 1,722) | 260 (8 - 6,414) | **0.039** |
| BUN (mg/dL) | 12 (3 - 39) | 12 (3 - 139) | 0.16 |
| Creatinine (mg/dL) | 0.86 (0.20 - 2.20) | 0.98 (0.30 - 6.30) | 0.28 |

[1]Mean (Min - Max); n (%).

[2]Wilcoxon rank sum test; Pearson's Chi-squared test.

(Table H in S1 Text). The difference in hematological parameters was observed for DENV 1 and DENV 2 serotypes, with Platelets count being lower (123 vs. 93 10^9/L; p = 0.025), while higher Creatinine level in DENV 2 serotype (0.76 vs 1.10 mg/dL; p = 0.002) (Table I in S1 Text). Additionally, the hematological parameters in serotype positive group (n = 126) were compared as per WHO criteria of DWWS and DWOS. In this analysis WBC (p = 0.024), Platelets (p = 0.006) and Creatinine (p = 0.007) were found to be significant (Table J in S1 Text). As per WHO criteria, 261 were classified as DWOS, 39 with DWWS and 5 as SD. This makes a total of 305 patients, 43 did not classify in any of the above categories, but presented with other clinical presentations. The median levels of AST (p = 0.016) and ALT (p = 0.009) were significantly higher in patients with SD compared to those with NSD (Table 4). A proportion of 49.8% (152/305) patients had abnormal ALT levels (>40 U/L), and 51.4% (157/305) of patients had abnormal AST levels (>40 U/L). ALT levels >1000 U/L were observed in six patients, two had dengue shock syndrome and one of them had co-infection with hepatitis A virus. Renal dysfunction was also significantly higher in severe dengue cases in comparison to NSD with elevated BUN (p = 0.005) and creatinine (p < 0.001) (Table 4). The SD group (n = 5) presented with Dengue shock syndrome, Acute Kidney injury, Multiorgan dysfunction (MOD), Septic shock, and Dengue Hemorrhagic fever.

Thrombocytopenia was further categorized into grade 0 (>150,000), grade I (75,000 – 150,000), grade II (50,000 – 75,000), grade III (25,000–50,000), and grade IV (< 25,000). Notably, there were significant differences in thrombocytopenia with respect to the days of illness (p = 0.03) and hospitalization (p = 0.001) (Fig 4A and 4B). Increased Days of illness and hospitalization was also observed in patients with higher grade of thrombocytopenia. In addition, the dengue severity groups also showed statistically significant differences with the hospitalization (p = 0.001) as shown in Fig 4C. Additionally,

**Table 4. Clinical parameters and their significance among dengue severity.**

| Characteristic | Dengue with warning signs | Dengue without Warning Signs | Severe DENGUE | p-value[2] |
|---|---|---|---|---|
| | N = 39[1] | N = 261[1] | N = 5[1] | |
| Hematocrit | 37.8 (5.7) | 39.3 (5.9) | 33.6 (8.5) | 0.11 |
| WBC Count | 5.7 (2.5) | 5.5 (3.6) | 7.9 (4.1) | 0.12 |
| Lymphocyte Count | 40 (17) | 37 (19) | 18 (5) | **0.049** |
| Neutrophil Count | 47 (19) | 53 (20) | 75 (8) | **0.014** |
| Hemoglobin | 12.44 (1.86) | 12.91 (2.09) | 11.13 (1.94) | 0.086 |
| Platelet Count | 82 (75) | 125 (89) | 179 (103) | **0.001** |
| ALT | 239 (442) | 132 (244) | 1,929 (2,324) | **0.009** |
| AST | 428 (1,119) | 173 (234) | 1,599 (2,022) | **0.016** |
| BUN | 11 (10) | 11 (13) | 26 (10) | **0.005** |
| Creatinine | 0.92 (0.65) | 0.89 (0.28) | 3.58 (2.21) | **<0.001** |
| Days of Hospitalization | 2.64 (2.22) | 1.65 (2.86) | 5.40 (5.55) | **0.002** |

[1]Mean (SD).

[2]Kruskal-Wallis rank sum test.

the neutrophil-to-lymphocyte ratio (NLR) was also found to be significantly deranged (NLR = 4.59) in SD patients. NLR was found to be positively correlated with platelet count (S3A and S3B Fig).

### 3.5. Clinical diagnosis of dengue mixed serotypes

Of those PCR positive (n = 203), only 157 were subjected to the PCR serotyping assay, 46 samples were not processed further due to limitation of reagents. 126/157 were positive with any one serotype of Dengue, and 31 was non typeable. Of these 126 cases 30 (23.8%) were initially negative for NS1 but later found to be DENV PCR + . Among those, 21 had Dengue fever/ Dengue like illness/ viral fever without any other complication, while 9 had Dengue fever with some complications involving GI symptoms (n = 1), upper respiratory tract symptoms (n = 2), and other complications such as metabolic acidosis (n = 1), drug reaction(n = 1), pancytopenia(n = 1), UTI (n = 1), suspected Zika virus infection (n = 1) and connective tissue disorder(n = 1). Of those 31 Non typeable serotypes, 13 had a diagnosis of dengue fever, 9 had a diagnosis of fever and viral fever, 7 were diagnosed with enteric fever, while 2 were diagnosed with tonsilitis and community acquired pneumonia. , They were all positive for Pan Dengue PCR (Sheet B in S1 Appendix). In addition, mixed serotype infection included (n = 21) cases, in which DENV 1|DENV 2 was predominant (n = 20) and only 01 case was found with DENV 1|DENV 3 co-infection. Of these, 13/21 were in-patients, while the remaining 8 were out-patients (Table 5).

Among the mixed serotype cases, 20 individuals were diagnosed with dengue fever/fever/suspected dengue fever/viral fever and 1 case was diagnosed with upper respiratory tract infection (Sheet A in S1 Appendix). Moreover, within mixed serotypes, a significant majority of cases were categorized as dengue without warning signs (18/21) (Table K in S1 Text). Further to this, it has been observed that a majority of patients in DENV 2 and mixed serotypes were hospitalized patients (74%). Only 05 patients were identified as DENV 3 and all of them were in-patient as shown in Table 5 and S4 Fig.

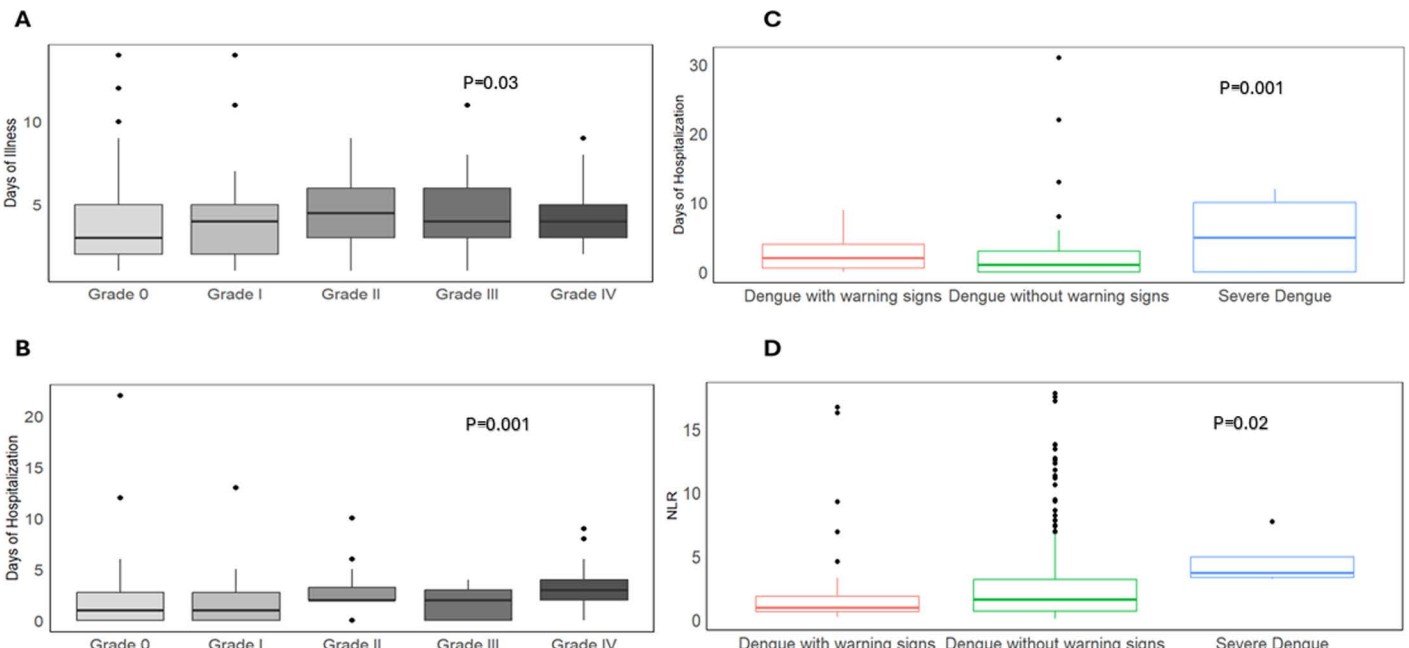

**Fig 4. Comparison of lab parameters with Disease severity.** Comparison of platelet grading and dengue severity groups with days of illness, hospitalization, and Neutrophil-to-Lymphocyte Ratio (NLR). (A) Association between days of illness and grade of platelet grading. (B) Association between hospitalization and platelet grading. (C) Association between dengue severity groups and hospitalization. (D) Association between dengue severity groups and NLR.

**Table 5. Clinical features of mixed serotypes.**

|  | In-patient (n = 13) N(%) | Out-patient (n = 8) N (%) |
|---|---|---|
| DV1/DV2 | 12 | 8 |
| DV1/DV3 | 1 | – |
| NS1+ | 12 (92) | 6 (75) |
| IgM + | 2 (15.3) | 1 (12.5) |
| DWWS | 3 (23) | 0 |
| DWOWS | 10 (77) | 8 (100) |
| Days of illness | 3.76 ± 1.92 | 3.75 ± 1.92 |
| Age | 44.76 ± 11.69 | 23.0 ± 10.15* |
| Lymphocytes | 33.0 ± 18.60 | 38.94 ± 12.35 |
| Neutrophils | 55.87 ± 20.67 | 48.98 ± 13.53 |
| Platelets | 87.46 ± 62.62 | 96.50 ± 6.691 |
| NLR ratio | 4.5 ± 6.91 | 1.50 ± 0.918 |
| Thrombocytopenia grade | Grade 0 (3) | Grade 0 (3) |
|  | Grade I (4) | Grade I (2) |
|  | Grade II (1) | Grade II (0) |
|  | Grade III (2) | Grade III (1) |
|  | Grade IV (3) | Grade IV (2) |

* p < 0.05.

## 4. Discussion

The first Dengue outbreak in Pakistan was reported in 1994, followed by significant outbreaks in 2005, 2011, 2017, 2018, 2019, and 2022 [8]. The recent outbreak reported 75,450 Dengue cases between January and November 2022 (Data from Pakistan's National Institute of Health) exacerbated by flooding. A shift in Dengue peak season and out of the season Dengue cases have also been seen in this region which is mainly due to climate change and increasing urbanization and heat waves. Such increase was predominantly noted for November and December 2021 where 16388 cases were reported [14]. Pakistan ranks 30th in the climate change performing index (CCPI) (https://ccpi.org/country/pak), indicating vulnerabilities to climate change impacts and, most importantly rampant increase in vector-borne diseases. Our study focused on the epidemiological characteristics of dengue virus infection among hospitalized and out-patient cohorts in Karachi, emphasizing a sudden rise in cases observed in 2022 [9,15], and unique clinical presentations of Dengue fever seen in hospitalized patients. In this study, most cases were enrolled between September – October 2022, a period marked by heavy rainfall and catastrophic flooding [4]. Moreover, August 2022 stands out as the wettest month in the past 50 years, and October 2022 had the highest number of dengue cases reported in Pakistan. The correlation between high precipitation levels and dengue incidence was evident, with flood-affected regions like Sindh experiencing a higher prevalence of infections [1,15,16]. Urban areas in Sindh province like Karachi, with high population density, closely packed communities, inadequate sewage or drainage systems, and open water storage due to severe water shortages, create ideal conditions for mosquito breeding and the transmission of the dengue virus [17,18].

In our study, we found a higher number of males being affected which was expected due to outdoor occupation and gender differences in health-seeking behavior, influenced by social and cultural factors [19]. Adults aged 16–45 years were most frequently affected, consistent with previous studies as being more vulnerable to viral illness which might be due to their increased exposure and outdoor activities [11,20].

Serotype analysis identified DENV 1 and DENV 2 as primary serotypes which is consistent with previous reports from Pakistan [19,21] and neighboring countries [22]. Notably, the 2005 & 2006 outbreaks in Karachi [23,24] and New Delhi were dominated by all 4 serotypes and predominantly DENV 3, which resulted in numerous fatalities [25]. The predominance of DENV 3 was also noted in Bangladesh for the cases screened between 2000–2024 [26].

Serotype DV1 was associated with DHF in Indonesia [27] with no significant change in hematological parameters except for raised lymphocyte count in DV1 patients. While serotype DV2 was shown to be associated with disease severity and had an increased risk of disease with warning signs [28,29]. In our study, DV2 was associated with thrombocytopenia which was also observed in sero-positive patients with warning signs (Tables I and J in S1 Text). Although co-infection with DV1/DV2 has been reported to be associated with higher grading of thrombocytopenia previously [5], no such association was observed in this study with 21 mixed infections with range and frequency of thrombocytopenia (43% had Grade II to Grade IV) (Table 5).

Most DENV cases in our study presented with fever, nausea or vomiting, body ache, and headache as major symptoms, while retro orbital pain and maculopapular rash were less common [ 15]. Details of clinical presentations are given in S1 Appendix. No significant differences in vitals were observed between PCR-positive and negative patients, except for raised body temperatures in PCR-positive cases, presumably due to more viral inflammation.

The hematological parameters of patients with and without DENV infection were within normal ranges, except for deranged platelet count, ALT, and AST levels. These abnormalities were more pronounced in SD patients. Elevated ALT and AST levels in both DENV-positive and negative cases were consistent with previous reports from Pakistan, indicating liver involvement during dengue infection [30]. Impaired liver function tests were frequently observed during dengue infection, particularly in patients with dengue hemorrhagic fever (DHF). In our study, less severe cases also showed deranged ALT and AST levels (>40IU/ml), and only one-fourth of the study participants had ALT within normal ranges [31]. The percentage of lymphocytes at the time of admission is highly suggestive of longer hospital stays and days of illness [32]. Among markers of inflammation, both lesser NLR and PLR ratios are suggestive of disease severity in Dengue and

previously used to distinguish Dengue from COVID-19 disease [33,34]. However, we found a significant positive correlation between Platelets-Lymphocyte Ratio (PLR) and NLR but no association of NLR was observed with disease severity, probably due to limited severe cases (S3A and S3B Fig).

The limitations of the current study are: 1) the non-availability of SD cases during our study enrollment period, possibly due to the decline of consent due to the unstable condition of patients; 2) lack of sequencing for serotype-negative samples; and, 3) serological confirmation for secondary Dengue infections. Despite these limitations, we present here a detailed report of Dengue presentations highlighting its overlap with other febrile illnesses such as gastric/enteric fever, upper respiratory tract infection (URTI), Urinary tract infection (UTI), acute kidney injury (AKI) and other viral-like illnesses. Of note, all of these patients were clinically managed as viral fever in the absence of lab diagnosis (Sheet B in S1 Appendix). The clinical presentation of Dengue is evolving over time, but with the limited sensitivity of current RDTs, lab confirmation of these patients was much improved by Dengue qRT-PCR. From a public health perspective, our study highlights a critical need for molecular diagnostics in all sentinel labs for effective clinical decision-making, outbreak response management and improved case detection. We also highlight the need for strong clinical suspicion of dengue in NS1-negative and IgM-negative cases, which were later confirmed by qRT-PCR. This underscores the importance of molecular detection, as the reliance on conventional diagnostics may lead to missed diagnoses, ultimately impacting patient management and clinical decision-making.

In summary, the current study conducted in a large tertiary care center in Pakistan highlights the profound impact of Dengue worsened by flooding and poor sanitation, posing considerable challenges to public health. The high prevalence of Dengue cases in flood-prone regions emphasizes the urgent need to address the impact of climate change on vector-borne diseases. Furthermore, the rising Dengue rates in urban centers like Karachi emphasize the importance of robust Dengue surveillance and control programs at the provincial and district levels including Karachi and its outskirts mainly deprived of basic health facilities.

## Supporting information

**S1 Fig. Serotype distribution of Dengue infection in Dengue positive and negative samples.** Dengue samples were subjected to PCR and serotyping using ZCD assay. (A) Out of a total of 283 samples, 203 (72%) were diagnosed as DENV+ and 80 (28%) as DENV− (B) Dengue serotyping of 157 DENV+ samples revealed a predominant classification into DENV 1 (n=43), DENV 2 (n=57), and DENV 3 (n=5) with mixed serotypes (n=21) [DENV 1|DENV 2 (n=20) and DENV 1|DENV 3 (n=1)] detected in 13% of cases.
(TIF)

**S2 Fig. Comparison of Ct values of identified DENV serotypes.** q-RTPCR Ct Value distribution in dengue serotypes. No significant differences in Ct values were observed between DENV1, DENV2, DENV3 and mixed serotypes.
(TIF)

**S3 Fig. Association of Neutrophil/Lymphocyte Ratio (NLR) and Platelets/Lymphocyte ratio (PLR).** NLR and PLR show a high positive correlation in patients with Dengue, (B) NLR and platelets count in pateints with Dengue severity.
(TIF)

**S4 Fig. Hospitalization status of study subjects with DENV serotypes.** Hospitalization status of Serotype-specific participants. High percentages of DENV2, mixed serotypes and non-subtype participants were in-patient.
(TIF)

**S1 Text. Table A.** ZCD assay primers for Dengue detection: Primer sequence for the assay used for DENV detection. **Table B.** ZCD assay probes for Dengue detection: Probe information for the assay used for DENV detection. **Table C.** Dengue serotyping specific probes in real time qRT-PCR: Primer sequence of four serotypes **Table D.** Lab Diagnosis

of samples using three diagnostic tests: Total number of cases positive with NS1, IgM and PCR with clinical diagnosis. **Table E.** Counts of DENV qRT-PCR with reference to Clinical Diagnosis: 2x2 table for sensitivity and specificity calculation for all qRT-PCR positive and negative cases in comparison to clinical diagnosis (Dx). Clinical Dx+ includes all dengue fever, dengue-like illness, suspected dengue fever, suspected viral fever. Clinical Dx- includes other than dengue/viral fever and with other comorbidities as mentioned in Sheet B in S1 Appendix. **Table F.** Counts of Dengue qRT-PCR with reference to Gold Standard IgM+/NS1+ : Count of DENV qRT-PCR with Gold Standard Diagnosis (IgM+/NS1+). **Table G.** Sensitivity of Clinical Diagnosis of DENV with Gold Standard IgM+/NS1+: Sensitivity and Specificity table for DENV qRT-PCR with Gold Standard Diagnosis. **Table H.** Significance of laboratory parameters among different serotypes: Comparison of lab parameters for DV1(n = 43), DV2(n = 57), DV3(n = 5) and Mixed serotypes (n = 21), Table shows Mean (SD) for all parameters. Kruskal Wallis test was applied for significant difference (p < 0.05). **Table I.** Significance of laboratory parameters among DV1 & DV2 serotypes: Comparison of lab parameters for DV1(n = 43), DV2(n = 57). Table shows Mean (SD) for all parameters. Wilcoxon Rank Sum test was applied for significant difference (p < 0.05). **Table J.** Dengue disease severity in Serotype positive samples (n = 126): Table compares the lab parameters of DENV serotype positive cases with and without Dengue Warning Signs. Kruskal Wallis test was applied for significant difference (p < 0.05). **Table K.** Clinical diagnosis of dengue serotypes (n = 126); All serotype positive cases are shown with and without Dengue Warning signs.
(DOCX)

**S1 Appendix. Excel sheet contain raw data of manuscript of 349 Patients with details of clinical and diagnostic tests in Sheet A, 41 cases with NS negative diagnosis in Sheet B, 21 mixed serotype cases in Sheet C, DENV positive and negative cases in Sheet D.**
(XLSX)

## Acknowledgments

Field staff, data management unit staff, patients and families for their contribution, BEI resource for provision of positive controls for PCR, Scott Weaver UTMB WRCEVA for the provision of positive controls.

## Author contributions

**Conceptualization:** Najeeha Talat Iqbal.

**Data curation:** Kumail Ahmed, Aqsa Khalid.

**Formal analysis:** Najeeha Talat Iqbal, Kumail Ahmed, Aqsa Khalid, Qamreen Mumtaz Ali.

**Funding acquisition:** Najeeha Talat Iqbal, Peter Rabinowitz, Wes C. Van Voorhis.

**Investigation:** Najeeha Talat Iqbal, Badar Afzal, Erum Khan, Peter Rabinowitz, Wes C. Van Voorhis.

**Methodology:** Najeeha Talat Iqbal, Kumail Ahmed, Khekashan Imtiaz.

**Project administration:** Tania Munir, Helene McOwen.

**Resources:** Jesse J. Waggoner, Helene McOwen.

**Supervision:** Hannah Fenelon, Peter Rabinowitz, Wes C. Van Voorhis.

**Visualization:** Aqsa Khalid.

**Writing – original draft:** Najeeha Talat Iqbal, Aqsa Khalid.

**Writing – review & editing:** Najeeha Talat Iqbal, Syed Faisal Mahmood, Unab Khan, Farah Naz Qamar, Jesse J. Waggoner, Erum Khan, Peter Rabinowitz, Wes C. Van Voorhis.

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
