## [Decision Letter · Decision Letter 0]

15 Jan 2025

PNTD-D-24-01513

Clinical characteristics and serotype association of dengue and dengue like illness in Pakistan

Dear Dr. Iqbal,

Thank you for submitting your manuscript to PLOS Neglected Tropical Diseases. After careful consideration, we feel that it has merit but does not fully meet PLOS Neglected Tropical Diseases's publication criteria as it currently stands. Therefore, we invite you to submit a revised version of the manuscript that addresses the points raised during the review process.

Please submit your revised manuscript within 60 days Mar 16 2025 11:59PM. If you will need more time than this to complete your revisions, please reply to this message or contact the journal office at plosntds@plos.org. Please include the following items when submitting your revised manuscript:

We look forward to receiving your revised manuscript.

Kind regards,

Edda Sciutto

Academic Editor

Paul Brindley

Editor-in-Chief

Shaden Kamhawi

co-Editor-in-Chief

Paul Brindley

co-Editor-in-Chief

**Additional Editor Comments (if provided):**

After reviewed the article entitled "Clinical characteristics and association of dengue serotypes and dengue-like diseases in

Pakistan", we found that the study is well conducted and presents the situation of dengue in patients treated in a tertiary care center in Pakistan.

Although we consider these findings potentially publishable in Plos of Neglected Diseases, the authors must first address the recommendations particularly made by the second reviewer.

Please submit the article including these recommendations, answering reviewer #2's comments one by one to be reviewed.

**Journal Requirements:**

**Reviewers' Comments:**

Reviewer's Responses to Questions

**Key Review Criteria Required for Acceptance?**

**Methods**

-Are the objectives of the study clearly articulated with a clear testable hypothesis stated?

-Is the study design appropriate to address the stated objectives?

-Is the population clearly described and appropriate for the hypothesis being tested?

-Is the sample size sufficient to ensure adequate power to address the hypothesis being tested?

-Were correct statistical analysis used to support conclusions?

-Are there concerns about ethical or regulatory requirements being met?

Reviewer #1: the objectives of the study clearly articulated with a clear testable hypothesis, and the study design appropriate to address the stated objectives. the sample size sufficient to ensure adequate power to address the hypothesis

Reviewer #2: A minimum sample size should be defined using statistical methods, this was ommitted.

Statistical methods to assess sensitivity and sensibility were not included, which was the gold standard? A detail statistical methodology must be included, and a valid scientific argument of using the selected gold standard according to the nature of the diagnosis of dengue.

**Results**

-Does the analysis presented match the analysis plan?

-Are the results clearly and completely presented?

-Are the figures (Tables, Images) of sufficient quality for clarity?

Reviewer #1: the results are clearly and completely presented

Reviewer #2: The title 3.2 sensitivity of serological pattern has no relationship with the presented results, as in the results described the sensitivity of clinical diagnosis compared to PCR.

3.5 Clinical Diagnosis of Dengue Mixed Serotypes: Is it possible to present this results in a table, for a better understanding of the data

Figures are not of sufficient quality

**Conclusions**

-Are the conclusions supported by the data presented?

-Are the limitations of analysis clearly described?

-Do the authors discuss how these data can be helpful to advance our understanding of the topic under study?

-Is public health relevance addressed?

Reviewer #1: the current study conducted in a large tertiary care center in Pakistan, highlights the profound

impact of Dengue worsened by flooding and poor sanitation, posing considerable challenges to public

health. The high prevalence of Dengue cases emphasizes the urgent need to address the importance of robust infrastructure and sanitation. Immediate measures by authorities are

crucial to control vector populations and protect public health. The main limitation of this study includes

lower number of dengue severe cases and the unavailability of the resources to perform dengue

sequencing for mixed infections.

Reviewer #2: Do the authors discuss how these data can be helpful to advance our understanding of the topic under study? The results and the data obtained, were no use correctly to discuss it

-Is public health relevance addressed? It should be expanded and improved in the discussion

**Editorial and Data Presentation Modifications?**

Reviewer #1: Accept

Reviewer #2: (No Response)

**Summary and General Comments**

Reviewer #1: The current study aims to explore the clinical presentations and features of dengue fever in a tertiary care hospital.

Collected data on cases including clinical symptoms and laboratory results including qRT-PCR and serotype characterization. The majority of subjects enrolled (75%) had mild disease without warning signs. The sensitivity

of clinical diagnosis was found to be 87.25% and the specificity of 68.35%. qRT-PCR detected 43.5% of cases

with viral fever initially screened negative. Screening with rapid tests requires further confirmation by molecular assay in cases with dengue and dengue-like illness.

Reviewer #2: A major revision is needed. The main concern is how the sensitivity and specificity results are presented, a rationale for the use of the gold standard is not detailed, and the methodology is not detailed. The discussion is brief, the data was used very little.The discussion should be expanded; nothing is discussed about the clinical findings and the serotypes found.

PLOS authors have the option to publish the peer review history of their article (what does this mean? ). If published, this will include your full peer review and any attached files.

**Do you want your identity to be public for this peer review?** For information about this choice, including consent withdrawal, please see our Privacy Policy .

Reviewer #1: No

Reviewer #2: No

**Figure resubmission:**
---

## [Editor Report · Decision Letter 1]

12 Mar 2025

Dear Iqbal,

We are pleased to inform you that your manuscript 'Clinical characteristics and serotype association of dengue and dengue like illness in Pakistan' has been provisionally accepted for publication in PLOS Neglected Tropical Diseases.

Best regards,

Edda Sciutto

Academic Editor

Paul Brindley

Editor-in-Chief

Shaden Kamhawi

co-Editor-in-Chief

Paul Brindley

co-Editor-in-Chief

In this new version, the authors have addressed and included the clarifications and modifications requested by the reviewers. The article is now acceptable for publication.

---

## [Editor Report · Acceptance letter]

Dear Iqbal,

We are delighted to inform you that your manuscript, "Clinical characteristics and serotype association of dengue and dengue like illness in Pakistan," has been formally accepted for publication in PLOS Neglected Tropical Diseases.

Best regards,

Shaden Kamhawi

co-Editor-in-Chief

Paul Brindley

co-Editor-in-Chief
